# Thermal Behavior of Green Cellulose-Filled Thermoplastic Elastomer Polymer Blends

**DOI:** 10.3390/molecules25061279

**Published:** 2020-03-12

**Authors:** Stefan Cichosz, Anna Masek

**Affiliations:** Institute of Polymer and Dye Technology, Faculty of Chemistry, Lodz University of Technology, Stefanowskiego 12/16, 90-924 Lodz, Poland; stefan.cichosz@dokt.p.lodz.pl

**Keywords:** cellulose fibers, ethylene-norbornene copolymer, thermogravimetric analysis, differential scanning calorimetry

## Abstract

A recently developed cellulose hybrid chemical treatment consists of two steps: solvent exchange (with ethanol or hexane) and chemical grafting of maleic anhydride (MA) on the surface of fibers. It induces a significant decrease in cellulose moisture content and causes some changes in the thermal resistance of analyzed blend samples, as well as surface properties. The thermal characteristics of ethylene-norbornene copolymer (TOPAS) blends filled with hybrid chemically modified cellulose fibers (UFC100) have been widely described on the basis of differential scanning calorimetry and thermogravimetric analysis. Higher thermal stability is observed for the materials filled with the fibers which were dried before any of the treatments carried out. Dried cellulose filled samples start to degrade at approximately 330 °C while undried UFC100 specimens begin to degrade around 320 °C. Interestingly, the most elevated thermal resistance was detected for samples filled with cellulose altered only with solvents (both ethanol and hexane). In order to support the supposed thermal resistance trends of prepared blend materials, apparent activation energies assigned to cellulose degradation (E_A1_) and polymer matrix decomposition (E_A2_) have been calculated and presented in the article. It may be evidenced that apparent activation energies assigned to the first decomposition step are higher in case of the systems filled with UFC100 dried prior to the modification process. Moreover, the results have been enriched using surface free energy analysis of the polymer blends. The surface free energy polar part (Ep) raises considering samples filled with not dried UFC100. On the other hand, when cellulose fibers are dried prior to the modification process, then the blend sample’s dispersive part of surface free energy is increased with respect to that containing unmodified fiber. As polymer blend Ep exhibits higher values reflecting enhanced material degradation potential, the cellulose fibers employment leads to more eco-friendly production and responsible waste management. This is in accordance with the rules of sustainable development.

## 1. Introduction

Constant development in the various branches of industry generates the need for more advanced materials, which are not only of improved performance, but also follow the rules of sustainability [1,2,3,4]. Therefore, the development of a new generation of eco-friendly materials representing a less environment-harmful alternative to commonly used plastics is highly desired [5]. As a consequence, more and more natural additives are being introduced to the polymer processing processes, e.g., various wood components [6,7,8,9,10], natural anti-aging agents [11,12,13], mineral fillers [14,15,16], starch-based materials [17,18].

Cellulose is a biopolymer consisting of glucose repeat units combined into polymer chains [19]. When introduced into a polymer matrix, cellulose may increase the blend strength and stiffness. Cellulose is of a low price, low density, and exhibits depressed CO_2_ emission during the thermal degradation process. In addition, natural fibers are annually renewable and fully biodegradable [20,21,22,23] important for sustainable development.

One of the biggest difficulties to overcome for the use of natural fiber filled polymer blends is their poor thermal stability [21]. Cellulose degrades at the temperature in the region of 350 °C and its surface can be easily hydrophobized [20]. A modifiable surface is extremely important to enhance the adhesion improvement between the non-polar polymer matrix and hydrophilic cellulose, whereby efficiency of a stress transfer and interfacial properties may be improved [24]. Therefore, cellulose seems to be a sufficient and promising polymeric filler blend reinforcing in comparison with other wood components:hemicellulose—degrades at around 300 °C and it may be easily hydrolyzed due to its random amorphous structure [25],lignin—composed of three different kinds of benzene-propane units which are dense crosslinked, exhibits a very high molecular weight and an increased thermal stability, very hard to decompose; nevertheless, lignin is not a major wood component [26],extractives—compounds of a lower molecular weight which decompose at relatively low temperature; may promote ignitability of wood at lower temperatures (consequence of high volatility) and accelerate the degradation process (the degradation of one component may cause an earlier decomposition of other wood compounds) [27].

Furthermore, cellulose contains crystalline regions which contribute to the thermal stability of the discussed biopolymer—hemicellulose and lignin are not crystalline [20]. It should be emphasized that the degree of cellulose crystallinity is one of the most important structural parameters responsible for material rigidity and thermal stability [28]. The free hydroxyl groups present in the cellulose macromolecules are likely to be involved in a number of intramolecular/intermolecular hydrogen bonds. This may promote the formation of various ordered molecular arrangements creation [19,21,29]. With increasing ratio of crystalline to amorphous regions, the flexibility of the macromolecules and the accessibility of hydroxyl moieties decreases [30]. Therefore, degradation processes are restricted [31].

Another interesting process for cellulose thermal stability improvement is hornification which relies on carrying out wetting-drying cycles of the biopolymer [32]. Some changes in the chemical structure of the natural fiber occur [33,34]. This allows a greater dimensional stability and a lower degradation via increasing the molecular packing [35]. Hornification is commonly carried out in the water environment and the whole process may be controlled by—e.g., number of cycles, drying parameters, etc. [35,36].

For this work hexane or ethanol as a non-polar and polar environment is employed. Only one wetting-drying cycle was carried out. Then, it was combined with a chemical modification which involves grafting of maleic anhydride (MA) on the surface of cellulose fibers via the reaction with accessible hydroxyl groups. The aim of the performed modification process was to reduce the moisture content in the cellulose fibers, improve their thermal resistance, and enhance the polymer matrix-filler interface in blend applications. 

The thermal properties of hybrid chemically modified cellulose fibers filled with ethylene-norbornene copolymer (TOPAS) blends have been studied. Moreover, surface property analysis which compares the polar and dispersive part of blend surface free energy depending on the performed modification process have been carried out. In this article, a new and innovative cellulose treatment procedure is introduced.

## 2. Materials and Methods

### 2.1. Preparation of the Specimens

#### 2.1.1. Materials

The type of cellulose used in this research was The Arbocel^®^ UFC100 Ultrafine Cellulose for Paper and Board Coating from *J*. *Rettenmaier* & Soehne (Rosenberg, Germany). Its density is about 1.3 g/cm^3^. It is insoluble in water and fats. Its average fiber length is about 8 μm. pH value varies between 5–7.5.

Given cellulose fibers have been modified with maleic anhydride (MA) obtained from Sigma-Aldrich^®^ (Darmstadt, Germany). It is a white solid substance with a molecular mass of 98.06 g/mol. In water it forms maleic acid. Its melting point is somewhere in the region of 51–56 °C; initial boiling point, 200 °C; and density 1.48 g/cm^3^ at 20 °C. The reagent was a commercial product of the highest purity available.

Solvents employed in the experiments, such as acetone (A), ethanol—99.9% solution (E) and hexane (H), were bought from Chempur^®^ (Piekary Śląskie, Poland). All of them are colorless substances and their properties are listed in the Table 1.

As a polymer matrix thermoplastic elastomer, ethylene-norbornene copolymer (TOPAS^®^ Elastomer E-140 from TOPAS Advanced Polymers^®^, Raunheim, Germany) was employed. This material is a high-performance alternative to traditional flexible materials for use in a broad range of applications, such as medical devices, injection molding articles for the optical industry, the packaging Industry. For processing, the crucial aspects are: the melting temperature, reported to be 84 °C; and the Vicat softening temperature, determined as 64 °C. The bulk density of the material is in between 450–550 g/dm^3^. Figure 1 reveals the structure of the ethylene-norbornene copolymer.

#### 2.1.2. Cellulose Fibers Hybrid Chemical Modification

In this research a new hybrid chemical treatment approach has been proposed. The regular surface modification of cellulose fibers was broadened with the solvent exchange in the filler, from water, to ethanol either hexane. This approach was adapted from another research carried out by Vuoti et al. [37] which considered paper-making applications.

Solvent exchange was performed before and after the surface modification of cellulose fibers with MA in order to observe its effect on, subsequently, chemical modification of cellulose fibers and properties of polymer blend samples. Moreover, solvents of different polarity employment influence have been examined. On the basis of sorption experiments ethanol was chosen as a polar solvent and as a non-polar one—hexane. They may exhibit different interactions with the natural filler. Furthermore, cellulose fibers have been dried (24 h, 100 °C; crystallizer 70 × 40 mm) either not dried before the hybrid chemical modification process in order to observe the effect of moisture content in the filler on the treatment yield. Additionally, some samples have been modified only with the solvent so as to examine the ethanol/hexane impact on the cellulose properties.

Solvent exchange: cellulose fibers were put into the flask and the solvent was poured (cellulose to solvent ratio—1:10). Then, such prepared dispersion has been mixed with the dipole (400 rpm for ethanol and 1000 rpm for hexane) in a room temperature. After 8 h, the mixing was switched off and the dispersion has been left for next 16 h in ambient conditions. When the time has been over, the solvent was distilled in a vacuum rotary evaporator at 40 °C (60 rpm, 100 mbar in case of ethanol, and 250 mbar for hexane).

Modification with maleic anhydride: cellulose was put into the acetone and MA solution (cellulose to acetone ratio—1:10, cellulose to MA ratio—4:1) for 2 h (oil bath 40 °C, 60 r/min) in a rotary evaporator. When the process of stirring came to an end, acetone was removed with the vacuum distillation process (oil bath 40 °C, initial pressure 200 mbar). Then, the sample was been subjected to the heat in a vacuum oven at 100 °C at 440 mbar for 4 h.

Between the steps of solvent exchange and modification with MA, cellulose fibers have been stored in a Binder^®^ oven (Tuttlingen, Germany) at 70 °C (crystallizer 70 × 40 mm) and then after finishing whole modification process—at 40 °C (crystallizer 70 × 40 mm). The whole modification process and its influence on the cellulose properties is widely described in the previous article concentrating on this topic [38] and the short summary is presented in Table 2.

#### 2.1.3. Polymer Blend Samples Preparation

Cellulose fibers were dried for 24 h at 100 °C (Binder^®^ oven; crystallizer 70 × 40 mm) before being incorporated into TOPAS. Then, polymer matrix (86 wt %) and cellulose fibers (14 wt %) have been mixed in a micromixer (Brabender Lab-Station from Plasti-Corder (Duisburg, Germany) with Julabo cooling system) at 110 °C for 30 min (50 rpm). Next, such prepared material has been put between two rolling mills prior to orientate fibers. Prepared mixture has been plasticized at 100 °C for 30 min in an oven and then put between two roll mills with 100 × 200 mm rolls, at a roll’s temperature of 20–25 °C and friction of 1:1.1 for approximately 45 s. The last step was to compress the blend plates between two steel molds, between two Teflon sheets, in a hydraulic press at 160 °C (electrically heated platens) for 10 min at approximately 125 bar. 

### 2.2. Polymer Blend Samples Characterization

#### 2.2.1. Thermogravimetric Analysis (TGA)

Thermogravimetric analysis (TGA) has been used in order to get acquainted with thermal degradation process detecting the mass loss as a function of raising temperature in the range from 25–600 °C (heating rate: 10 °C/min; air 50 cm^3^/min). Mettler Toledo TGA/DSC 1 STARe System equipped with a Gas Controller GC10 (Greifensee, Switzerland) has been employed in this investigation. Apparent activation energy (E_A_) values for the following decomposition steps are calculated with the use of Broido’s method [39]
(1)$$y=\frac{m_{t}-m_{\infty }}{m_{0}-m_{\infty }}$$
(2)$$\ln\left[\ln\left(\frac{1}{y}\right)\right]=-\frac{E_{A}}{R}\cdot \frac{1}{T}+C\;as\;a\;linear\;function:Y=aX+b$$
(3)$$\mathrm{where}:Y=\ln\left[\ln\left(\frac{1}{y}\right)\right],\;X=\frac{1}{T},a=-\frac{E_{A}}{R},b=C$$
(4)$$therefore\;E_{A}=-a\cdot R$$
where:

$m_{t}$—specimen mass at the time *t* (g)

$m_{0}$—specimen mass at the beginning of considered decomposition step (g)

$m_{\infty }$—specimen mass at the end of considered decomposition step (g)

T—temperature (K)

R—gas constant (8.31 J/(mol·K))

#### 2.2.2. Differential Scanning Calorimetry (DSC)

Differential scanning calorimetry (DSC) investigation has been performed in a temperature range from −40–200 °C (heating rate: 10 °C/min; argon atmosphere) prior to analyzing changes in glass transition temperature of ethylene elastic segments (Tg_1_), glass transition temperature of rigid norbornene segments (Tg_2_), and softening enthalpy (ΔH). Here, as well, Mettler Toledo TGA/DSC 1 STARe System equipped with Gas Controller GC10 has been employed.

#### 2.2.3. Surface Free Energy (SFE) Determination

Surface free energy has been determined on the basis of contact angle measurements done for three liquids: distilled water, ethylene glycol, and 1,4-diiodomethane. Droplets had a volume of approximately 2 μL. Surface of polymer blends has been cleaned with the use of acetone before the contact angle measurements was done. OCA 15EC goniometer by DataPhysics Instruments GmbH^®^ (Filderstadt, Germany) equipped with single direct dosing system (0.01–1 mL B. Braun^®^ syringe, Hassen, Germany) was employed. Surface free energy is calculated thanks to the Owens–Wendt–Rabel–Kaelble (OWRK) method [40].
(5)$$E=E_{P}+E_{D}$$
(6)$$\frac{\sigma_{L}\left(1+\cos\Theta \right)}{2\sqrt{\sigma_{L}^{D}}}=\sqrt{\sigma_{S}^{P}}\cdot \sqrt{\frac{\sigma_{L}^{P}}{\sigma_{L}^{D}}}+\sqrt{\sigma_{S}^{D}}\;as\;a\;linear\;function:Y=a\cdot X+b$$
(7)$$while:\;Y=\frac{\sigma_{L}\left(1+\cos\Theta \right)}{2\sqrt{\sigma_{L}^{D}}},\;X=\sqrt{\frac{\sigma_{L}^{P}}{\sigma_{L}^{D}}},\;a=\sqrt{\sigma_{S}^{P}},\;b=\sqrt{\sigma_{S}^{D}}$$
(8)$$therefore\;E_{P}=a^{2}=\sigma_{S}^{P}\;and\;E_{D}=b^{2}=\sigma_{S}^{D}$$
where:

E—total surface free energy (mJ/m^2^)

$E_{P}$–polar part of surface free energy (mJ/m^2^)

$E_{D}$–dispersive part of surface free energy (mJ/m^2^)

$\sigma_{L}$—total liquid surface tension (mN/m)

$\sigma_{L}^{P},\sigma_{L}^{D}$—respectively: polar and dispersive part of liquid surface tension (mN/m)

$\sigma_{S}^{P},\;\sigma_{S}^{D}$—respectively: polar and dispersive part of solid surface tension (mN/m)

Θ—contact angle (◦)

## 3. Results and Discussion

### 3.1. TGA Characterization

Thermogravimetric analysis helps to assess the thermal decomposition of investigated polymer blends. In the Figure 2, a TGA curve typical for ethylene-norbornene copolymer filled with cellulose may be observed. It consists of two major decomposition steps. The first one is assigned to the biopolymer thermal degradation and the second one to the polymer matrix disintegration.

Considering Figure 2 and Figure 3, TGA curves of TOPAS filled with modified cellulose samples are presented. While comparing them between each other, it may be suspected that the thermal decomposition of analyzed specimens is similar in case of all investigated blends. Moreover, the most dynamic mass loss for investigated blend samples is detected between 300 °C and 500 °C, as it has been observed in case of different research studies [41]. Furthermore, around 360 °C cellulose usually exhibit a sharp decline with an overall mass loss of approximately 80% [42]. Nevertheless, according to data gathered in Table 3, some crucial changes might be noticed.

Firstly, filled systems exhibit lower T_05%_, which is considered to be the initial decomposition temperature, in comparison with the neat polymer matrix. In case of TOPAS it is 411 °C, while regarding modified fibers incorporation the value drops to, e.g., 317 °C (TOPAS + UFC100/ND/MA/0). A similar effect has been observed in other research study [43]. Moreover, this could be explained by the lower thermal stability of cellulose in comparison with the polymer matrix [44] which is a consequence of hydroxyl moieties presence [45]. 

Secondly, systems with the addition of UFC100/D exhibit higher T_05%_ than while UFC100/ND added, e.g., TOPAS + UFC100/ND/MA/1/H − T_05%_ = 321 °C, TOPAS + UFC100/D/MA/1/H − T_05%_ = 327 °C. Generally, in case of fibers dried prior to chemical treatment incorporation, T_05%_ varies between 318 °C and 335 °C and while UFC100 is not dried before the modification, the value is in the range from 315–332 °C.

What should be emphasized, grafted MA—which increases the filler-polymer matrix interface properties—at the same time may contribute to a decrease in thermal stability of cellulose-filled systems. The maleic anhydride presence could alter the polymer chains orientation in the investigated materials as close packing of polymer chains could be inhibited by grafted structures [46]. The lower thermal stability of MA treated fibers may be overcome with the solvent exchange step incorporation, e.g., TOPAS + UFC100/D/MA/0 − T_05%_ = 326 °C, TOPAS + UFC100/D/MA/2/E − T_05%_ = 332 °C. Yet, the highest thermal resistance is observed in case of a treatment which employs only ethanol either hexane, e.g., TOPAS + UFC100/D/1/E − T_05%_ = 335 °C, TOPAS + UFC100/ND/1/H − T_05%_ = 332 °C, TOPAS + UFC100/D/MA/2/E − T_05%_ = 332 °C. Therefore, an appropriate treatment adjusting cellulose properties is highly required [47]. 

Moreover, decomposition rate (Table 3) is varied the most as long as the biopolymer exist in a system. When cellulose has been already degraded, the decomposition process follows the same path considering all analyzed blends—from T_50%_ which is approximately 460 °C, e.g., TOPAS + UFC100/ND/MA/1/H − T_50%_ = 460 °C, TOPAS + UFC100/D/MA/1/H − T_50%_ = 458 °C. Yet, according to the data available in the literature the molecular weight of a polymer matrix which has a great impact on product thermal stability should be also considered [48].

In order to support the supposed thermal resistance trends of prepared blend materials, apparent activation energies assigned to cellulose degradation (E_A1_) and polymer matrix decomposition (E_A2_) have been calculated [39]. According to the values shown in Table 4, it may be once more evidenced that apparent activation energies assigned to the first decomposition step are higher in case of the systems filled with UFC100 dried prior to the modification process, e.g., TOPAS + UFC100/ND/MA/0 − E_A1_ = (167 ± 2) kJ/mol, TOPAS + UFC100/D/MA/0 − E_A1_ = (195 ± 2) kJ/mol. Moreover, the most elevated values are assigned to the fibers modified with the employment of only solvents and not MA, e.g., TOPAS + UFC100/ND/1/H − E_A1_ = (230 ± 2) kJ/mol, TOPAS + UFC100/D/1/E − E_A1_ = (194 ± 2) kJ/mol. 

Although, it was not visible well with the previously presented analysis of TGA curve, there are some major differences within the second decomposition step characteristics among analyzed blend specimens, as values of E_A2_ for systems filled with modified UFC100 varies in the range from 213–243 kJ/mol. According to data presented in Table 3, it is visible that while the decomposition step apparent activation energy increases, then the thermal degradation temperature raises. 

Elevated apparent activation energies, e.g., TOPAS + UFC100/ND/1/H − E_A1_ = (230 ± 2) kJ/mol, may be explained by the higher thermal resistance of the biofiller after the modification process [49] either the developed polymer matrix–cellulose interface [50]. Another explanation may be a hornification process which relies on carrying out wetting–drying cycles of the biopolymer [32] which are present during the carried out modification process. Subsequently, some changes in the chemical structure of the natural fiber might occur [33,34] not only due to the chemical grafting but also thanks to performed wetting–drying cycles with various solvents. This may affect dimensional stability (higher molecular packing) [35].

Summarizing, the highest thermal resistance of prepared blend material is noticed regarding the systems filled with dried and modified with solvents cellulose fibers. Nevertheless, ethanol either hexane employment in UFC100 modification process may undoubtedly improve thermal properties of prepared blends, e.g., TOPAS + UFC100/ND/MA/0 − E_A1_ = (167 ± 2) kJ/mol, TOPAS + UFC100/ND/MA/1/H − E_A1_ = (186 ± 3) kJ/mol. Solvent employment contributes to the increase of apparent activation energy assigned to the first decomposition step.

### 3.2. DSC Analysis

Differential scanning calorimetry (DSC) has been employed in order to gather data about the influence of the filler modification process on glass transition temperature of both ethylene (Tg_1_), as well norbornene (Tg_2_) segments. Moreover, the process of material softening was observed and its enthalpy change has been calculated (ΔH). 

In Figure 4 DSC curves of blends filled with neat cellulose fibers and the maleinized ones (dried or undried before the modification process) are presented. At first glance, it is visible that the TOPAS + UFC100/ND/MA/0 behaves differently than the specimens filled with dried natural fibers. Nevertheless, the curve shapes are similar to each other.

Moreover, according to data gathered in Figure 5a–d, it may be claimed that hybrid chemical modification does not change the shape of the DSC curves significantly whether the cellulose is dried either not dried before the performed treatment. Therefore, hybrid chemical modification proposed in this research is supposed not to have a great impact on the glass transition and softening process of the blend.

On the other hand, considering Figure 5e–f, the thermal effects of ongoing transitions are slightly different depending on the solvent employed. It is very interesting due to the fact that hexane and ethanol might interact differently with treated cellulose fibers [51,52]. What is more, if some solvent particles are adsorbed on the surface of the biopolymer, then they may cause varied interactions with the polymer matrix macromolecules [51]. Therefore, the thermal behavior could be altered.

Regarding the data gathered in Table 5, similar observations can be made. Generally, in comparison with the neat TOPAS, modified cellulose fibers incorporation results in lowering the ethylene segments glass transition temperature values (Tg_1_) in case of UFC100/ND employment and its improvement for filling the polymer matrix with UFC100/D. Furthermore, there is no significant impact of UFC100 incorporation on norbornene segment glass transition temperature (Tg_2_) as it has been observed in different studies [53].

Moreover, the softening of the material requires lesser energy in case of filled blends, e.g., TOPAS − ΔH = 54 J/g, TOPAS + UFC100/ND/MA/O − ΔH = 41 J/g, TOPAS + UFC100/D/MA/O − ΔH = 45 J/g; similar effect has been observed before [24,54]. However, there are no variations considering temperature of this transition (T_peak_) as it is noted in different research studies [50,55]. T_peak_ is approximately 90 °C in case of all performed modifications and the softening enthalpy values varies between 41 J/g and 47 J/g for fibers not dried before the carried out treatments and between 42 J/g and 45 J/g in case of the biopolymer dried prior to the hybrid chemical modification.

On the other hand, as mentioned earlier, a great impact of the UFC100 modification path on the Tg_1_ values has been detected. Fibers drying prior to the modification process improves the glass transition temperature of elastic ethylene segments, e.g., TOPAS + UFC100/ND/MA/0 − Tg_1_ = 4 °C and TOPAS + UFC100/D/MA/0 − Tg_1_ = 10 °C, TOPAS + UFC100/ND/MA/1/H − Tg_1_ = 4 °C and TOPAS + UFC100/D/MA/1/H − Tg_1_ = 11 °C. Furthermore, while the filler is treated only with the use of solvents, also an increase in Tg_1_ value is detected and its effect is similar no matter if the fibers were dried before the solvent exchange or not, e.g., TOPAS + UFC100/ND/1/H − Tg_1_ = 13 °C, TOPAS + UFC100/D/1/H − Tg_1_ = 11 °C. 

Moreover, considering the glass transition temperature of norbornene rigid segments, there is no significant impact of the cellulose fibers modification on the Tg_2_ value, e.g., TOPAS + UFC100/ND/MA/0 − Tg_2_ = 39 °C, TOPAS + UFC100/D/MA/0 − Tg_2_ = 37 °C, TOPAS + UFC100/ND/1/E − Tg_2_ = 36 °C, TOPAS + UFC100/D/1/E − Tg_2_ = 38 °C. What should be emphasized is that there is a powerful influence of the drying process on blends thermal behavior, but solvent exchange combined with MA modification does not have a great impact on analyzed thermal transitions. Yet, while only solvent treatment incorporated, the glass transition temperature of ethylene segments is elevated. What is more, the behavior of TOPAS + UFC100/D/1440 sample is almost the same as in case of TOPAS filled with modified fibers which were not dried prior to the modification process. 

### 3.3. SFE Determination

Among different modifications of cellulose fibers which have been performed, some variations of blend sample surface free energy were detected. Therefore, due to the performed research study, it may be claimed that not only the cellulose surface energy is altered upon the modification process [56,57], but also the blend samples surface energy varies [58]. In Table 6, the values of neat polymer matrix (TOPAS) surface free energy and its polar part are presented. Obtained values are similar to the results obtained for neat polyethylene [59] with slight variations which could be an effect of norbornene content.

Furthermore, according to Figure 6, one can easily observe the differences in water droplet behavior on the surface of analyzed blend samples. This is the direct consequence of surface free energy changes while TOPAS is filled with cellulose fibers. It is visible that the specimen shown in Figure 6b is wetted more easily in comparison with the neat polymer matrix.

Furthermore, according to the data gathered in Figure 7b, there are less variations observed regarding surface free energy values of analyzed blend samples. Nevertheless, E parameter of TOPAS + UFC100/D/MA/2/E is significantly improved. What should be also emphasized, there are no trends visible considering the solvent employment in the modification process considering the surface free energy polar part variations. In general, E_p_ is lower than in case of the same treatments performed in case of UFC100/ND. It may be said that while cellulose fibers are dried prior to the modification process, then the blend sample’s dispersive part of surface free energy increases.

What should be also emphasized, the wetting properties depends not only on total surface free energy, but they also vary regarding the polar and dispersive components [60]. Therefore, water would wet easier the surface which possess a higher value of surface energy polar part, as H_2_O is a highly polar solvent [61], e.g., water droplet shown in Figure 6 exhibit the lower value of contact angle as a consequence of higher surface energy polar part and not the changes in the total surface energy.

## 4. Conclusions

In this article, an effect of the newly incorporated hybrid chemical modification of cellulose fibers on thermal properties of polymer blends is presented. Regarding the thermal properties of analyzed blend samples, in general, higher thermal stability is obtained in case of the materials filled with the fibers which were dried before any of the treatments carried out. What is also interesting is that elevated thermal resistance was detected considering the samples filled with cellulose altered only with solvents (both ethanol and hexane) no matter if cellulose fibers were dried prior to chemical modification or not. It should be underlined that there is undoubtedly a powerful influence of the drying process on blends thermal behavior. Apparent activation energies assigned to the first decomposition step are higher in case of the systems filled with UFC100 dried prior to the modification process, e.g., TOPAS + UFC100/ND/MA/0 − E_A1_ = (167 ± 2) kJ/mol, TOPAS + UFC100/D/MA/0 − E_A1_ = (195 ± 2) kJ/mol. Moreover, the most elevated values are assigned to the fibers modified with the employment of only solvents and not MA, e.g., TOPAS + UFC100/ND/1/H − E_A1_ = (230 ± 2) kJ/mol, TOPAS + UFC100/D/1/E − E_A1_ = (194 ± 2) kJ/mol. What should be also mentioned, a significant impact of the treatment kind on glass transition temperature of elastic ethylene segments was noticed among all of performed modifications e.g., TOPAS + UFC100/ND/MA/0 − Tg_1_ = 4 °C and TOPAS + UFC100/D/MA/0 − Tg_1_ = 10 °C, TOPAS + UFC100/ND/MA/1/H − Tg_1_ = 4 °C and TOPAS + UFC100/D/MA/1/H − Tg_1_ = 11 °C. 

Considering surface properties, there are no trends visible considering the solvent employment in the modification process considering the surface free energy polar part variations. In general, E_p_ is lower than in the case of the same treatments performed in the case of UFC100/ND. However, it may be said that while cellulose fibers are dried prior to the modification process, then the blend sample’s dispersive part of surface free energy increases. Subsequently, when cellulose is not dried, the surface energy polar part of a specimen raises. 

In general, the solvent exchange process seems to enhance the biofiller thermal resistance. It is the phenomenon similar in its origin to the hornification. Nevertheless, solvents of different polarity may vary the properties of modified cellulose fibers. As UFC100 characteristics are improved, while TOPAS is loaded with treated natural filler, then the thermal stability of polymer blend sample also increases. 

Presented materials are extremely important regarding an opportunity of creation of bio-inspired polymeric blends for various applications, e.g., packaging, furniture, automotive, aviation, sport equipment, etc. Moreover, cellulose fibers cause an increase of material degradation potential (easier material wetting) which leads to more eco-friendly production and aware waste management fulfilling the rules of sustainable development. What should be taken into consideration is that changes in thermal behavior of cellulose-filled polymer blends (assessed with DSC and TGA) cannot be understood as increased degradation potential of material in the natural environment—this is the question of easier wetting.

## Figures and Tables

**Figure 1 molecules-25-01279-f001:**
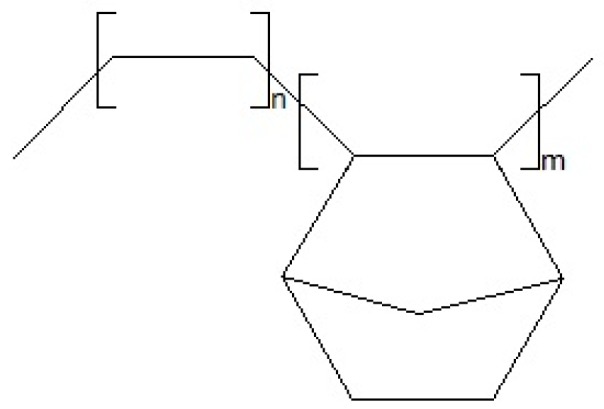
Ethylene-norbornene copolymer structure.

**Figure 2 molecules-25-01279-f002:**
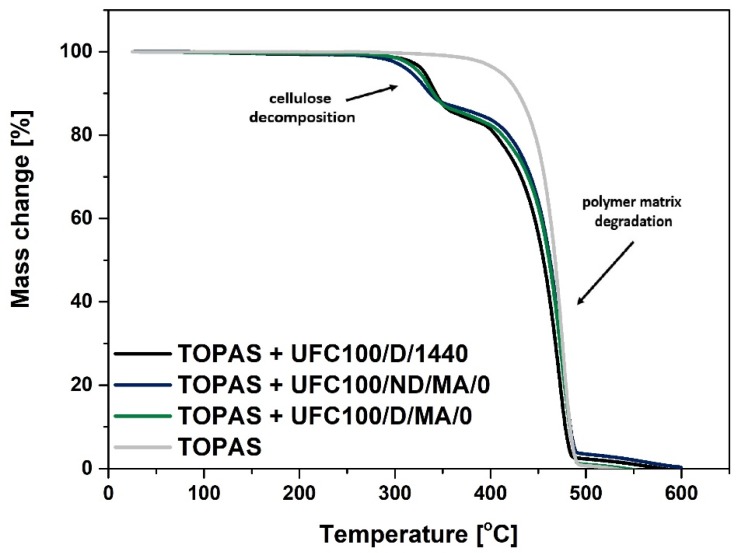
TGA curves of analyzed blend samples filled with MA treated cellulose fibers (regular chemical modification approach) dried either not dried before the carried out process; UFC100/D/1440—cellulose fibers dried for 1440 min (24 h) at 100 °C.

**Figure 3 molecules-25-01279-f003:**
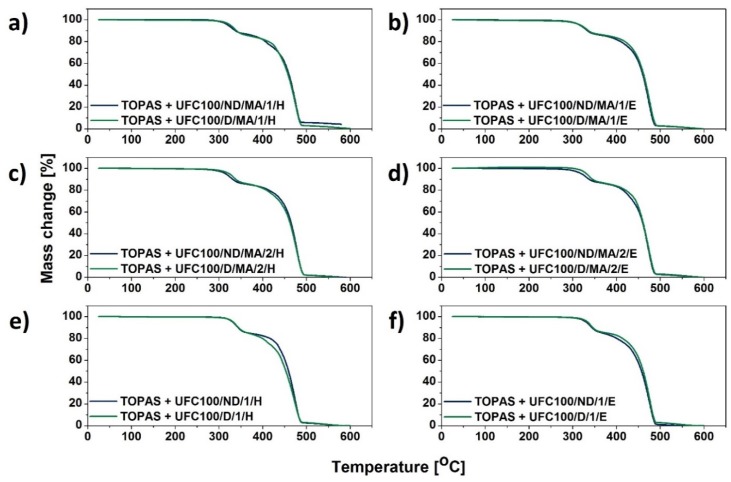
TGA curves of blend samples filled with cellulose fibers: (**a**) solvent exchanged to hexane and then MA treated; (**b**) solvent exchanged to ethanol and then MA treated; (**c**) MA treated and then solvent exchanged to hexane; (**d**) MA treated and then solvent exchanged to ethanol; *(***e**) solvent exchanged to hexane; (**f**) solvent exchanged to ethanol.

**Figure 4 molecules-25-01279-f004:**
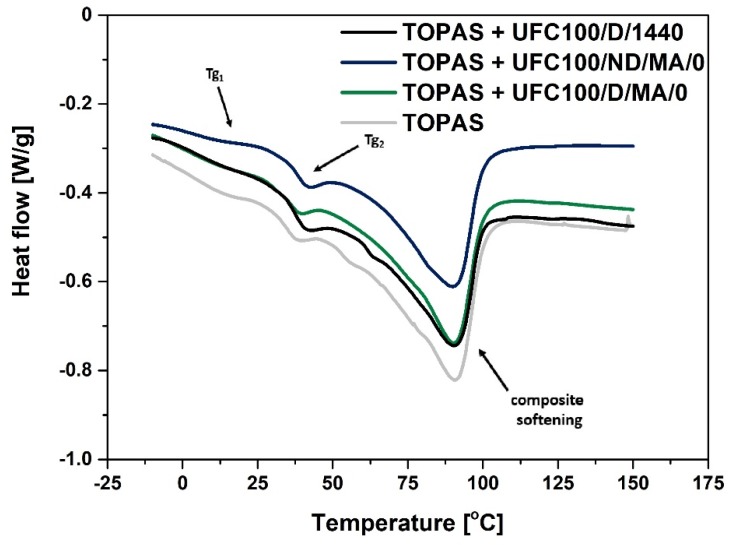
DSC curves of analyzed blends filled with MA treated cellulose fibers: Tg_1_—glass transition of ethylene segments; Tg_2_—glass transition of norbornene rings segments; UFC100/D/1440—cellulose fibers dried for 1440 min (24 h) at 100 °C.

**Figure 5 molecules-25-01279-f005:**
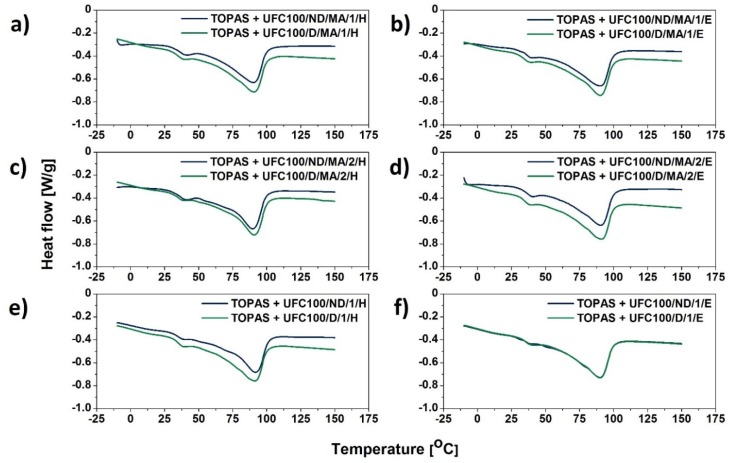
DSC curves of blend samples filled with cellulose fibers: (**a**) solvent exchanged to hexane and then MA treated; (**b**) solvent exchanged to ethanol and then MA treated; (**c**) MA treated and then solvent exchanged to hexane; (**d**) MA treated and then solvent exchanged to ethanol; (**e**) solvent exchanged to hexane; (**f**) solvent exchanged to ethanol.

**Figure 6 molecules-25-01279-f006:**
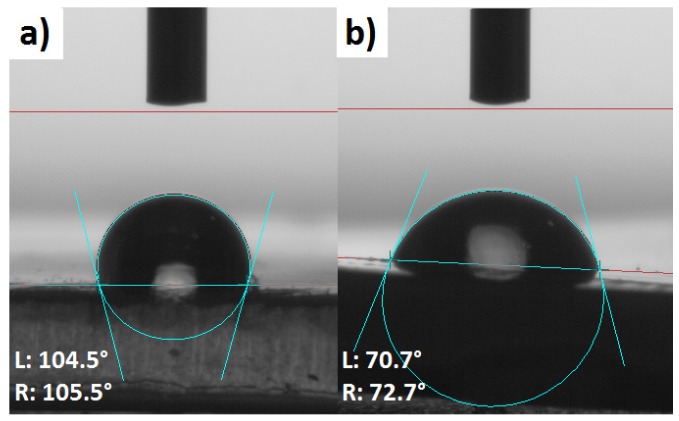
Distilled water droplet on the surface of blend samples: (**a**) TOPAS; (**b**) TOPAS + UFC100/D/MA/1/E.

**Figure 7 molecules-25-01279-f007:**
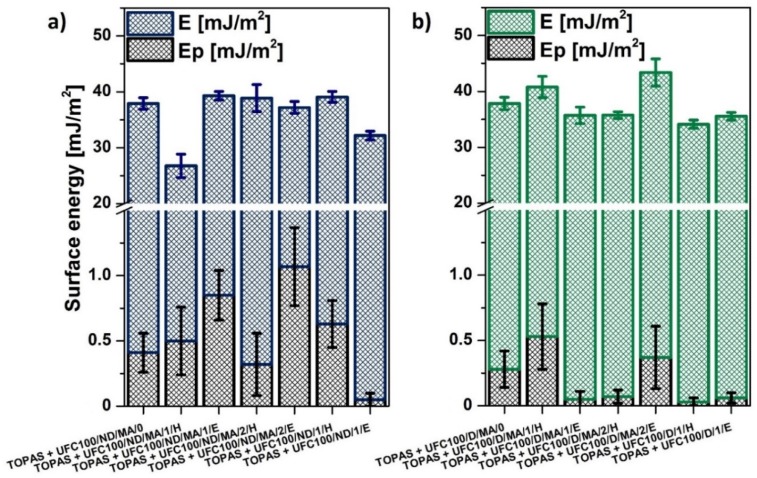
Surface free energy of investigated blends filled with modified cellulose fibers: (**a**) not dried before the modification process; (**b**) dried before the modification process.

**Table 1 molecules-25-01279-t001:** Physical properties of solvents employed in experiments.

Property	Acetone	Ethanol (99.9%)	Hexane
boiling point (°C)	55–57	78	68
viscosity at 20 °C (mPas)	0.330	1.078	0.310
density at 20 °C (g/cm^3^)	0.791	0.790	0.660

**Table 2 molecules-25-01279-t002:** Summary of all performed cellulose fibers modifications.

Sample	Dried Before Modification (D)	Not Dried Before Modification (ND)	Solvent Exchange				MA Treated
			Before MA Treatment		After MA Treatment		
			H	E	H	E	
UFC100/ND/MA/0	**------**	✔	**------**	**------**	**------**	**------**	✔
UFC100/ND/MA/1/H	**------**	✔	✔	**------**	**------**	**------**	✔
UFC100/ND/MA/1/E	**------**	✔	**------**	✔	**------**	**------**	✔
UFC100/ND/MA/2/H	**------**	✔	**------**	**------**	✔	**------**	✔
UFC100/ND/MA/2/E	**------**	✔	**------**	**------**	**------**	✔	✔
UFC100/ND/1/H	**------**	✔	✔	**------**	**------**	**------**	**------**
UFC100/ND/1/E	**------**	✔	**------**	✔	**------**	**------**	**------**
UFC100/D/MA/0	✔	**------**	**------**	**------**	**------**	**------**	✔
UFC100/D/MA/1/H	✔	**------**	✔	**------**	**------**	**------**	✔
UFC100/D/MA/1/E	✔	**------**	**------**	✔	**------**	**------**	✔
UFC100/D/MA/2/H	✔	**------**	**------**	**------**	✔	**------**	✔
UFC100/D/MA/2/E	✔	**------**	**------**	**------**	**------**	✔	✔
UFC100/D/1/H	✔	**------**	✔	**------**	**------**	**------**	**------**
UFC100/D/1/E	✔	**------**	**------**	✔	**------**	**------**	**------**

**Table 3 molecules-25-01279-t003:** Temperatures of the mass loss; T_X%_—temperature at which the mass loss of x% is detected.

Sample	T_05%_ (°C)	T_10%_ (°C)	T_15%_ (°C)	T_20%_ (°C)	T_50%_ (°C)	T_80%_ (°C)	T_90%_ (°C)
TOPAS	411	430	440	447	468	479	483
TOPAS + UFC100/D/1440	331	343	366	405	456	473	478
TOPAS + UFC100/ND/MA/0	317	337	387	419	462	478	484
TOPAS + UFC100/ND/MA/1/H	321	338	381	406	460	478	483
TOPAS + UFC100/ND/MA/1/E	317	333	375	410	460	477	482
TOPAS + UFC100/ND/MA/2/H	315	332	369	414	465	481	486
TOPAS + UFC100/ND/MA/2/E	320	338	385	415	460	476	481
TOPAS + UFC100/ND/1/H	332	344	367	417	459	476	481
TOPAS + UFC100/ND/1/E	330	344	368	401	457	476	482
TOPAS + UFC100/D/MA/0	326	340	374	412	462	478	484
TOPAS + UFC100/D/MA/1/H	327	340	372	413	458	477	482
TOPAS + UFC100/D/MA/1/E	318	336	384	420	462	478	484
TOPAS + UFC100/D/MA/2/H	324	339	375	409	462	480	486
TOPAS + UFC100/D/MA/2/E	332	345	389	421	461	477	483
TOPAS + UFC100/D/1/H	333	345	366	400	453	474	481
TOPAS + UFC100/D/1/E	335	346	377	414	461	477	483

**Table 4 molecules-25-01279-t004:** Tabularized values of apparent activation energy assigned to the decomposition steps calculated with the use of Broido’s method [39]; $E_{A_{1}}$—apparent activation energy of cellulose degradation, $E_{A_{2}}$—apparent activation energy of polymer matrix decomposition.

Sample	$E_{A_{1}}(\mathrm{kJ}/\mathrm{mol})$	$E_{A_{2}}(\mathrm{kJ}/\mathrm{mol})$
TOPAS	---------	225 ± 3
TOPAS + UFC100/D/1440	189 ± 2	232 ± 2
TOPAS + UFC100/ND/MA/0	167 ± 2	232 ± 3
TOPAS + UFC100/ND/MA/1/H	186 ± 3	218 ± 4
TOPAS + UFC100/ND/MA/1/E	174 ± 2	228 ± 3
TOPAS + UFC100/ND/MA/2/H	176 ± 2	224 ± 3
TOPAS + UFC100/ND/MA/2/E	172 ± 2	234 ± 3
TOPAS + UFC100/ND/1/H	230 ± 2	243 ± 2
TOPAS + UFC100/ND/1/E	193 ± 2	213 ± 3
TOPAS + UFC100/D/MA/0	195 ± 2	224 ± 3
TOPAS + UFC100/D/MA/1/H	195 ± 2	233 ± 2
TOPAS + UFC100/D/MA/1/E	184 ± 3	235 ± 3
TOPAS + UFC100/D/MA/2/H	183 ± 2	213 ± 3
TOPAS + UFC100/D/MA/2/E	194 ± 1	234 ± 3
TOPAS + UFC100/D/1/H	194 ± 2	217 ± 3
TOPAS + UFC100/D/1/E	194 ± 2	235 ± 3

**Table 5 molecules-25-01279-t005:** Tabularized maximum values of ethylene segments glass transition temperatures (Tg_1_), norbornene segments glass transition temperatures (Tg_2_), peak temperature (T_peak_) of the softening process and its enthalpy change (ΔH)

Sample	Tg_1_ (°C)	Tg_2_ (°C)	T_peak_ (°C)	ΔH (J/g)
TOPAS	9	36	90	54
TOPAS + UFC100/D/1440	4	38	90	40
TOPAS + UFC100/ND/MA/0	4	39	90	41
TOPAS + UFC100/ND/MA/1/H	4	39	90	42
TOPAS + UFC100/ND/MA/1/E	4	37	90	43
TOPAS + UFC100/ND/MA/2/H	4	38	90	41
TOPAS + UFC100/ND/MA/2/E	4	37	91	42
TOPAS + UFC100/ND/1/H	13	36	92	43
TOPAS + UFC100/ND/1/E	11	36	90	47
TOPAS + UFC100/D/MA/0	10	37	90	45
TOPAS + UFC100/D/MA/1/H	11	36	91	45
TOPAS + UFC100/D/MA/1/E	12	35	90	44
TOPAS + UFC100/D/MA/2/H	12	37	91	43
TOPAS + UFC100/D/MA/2/E	13	35	91	45
TOPAS + UFC100/D/1/H	11	35	90	44
TOPAS + UFC100/D/1/E	12	38	90	44

**Table 6 molecules-25-01279-t006:** Surface free energy (E) of neat polymer matrix and its polar component (E_p_).

E [mJ/m^2^]	E_p_ [mJ/m^2^]
40 ± 1	0.10 ± 0.06

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
