# Peer review of "Thermal Behavior of Green Cellulose-Filled Thermoplastic Elastomer Polymer Blends"

_molecules, 2020, doi:10.3390/molecules25061279_

Round 1

Reviewer 1 Report

The presented work is of scientific interest, and touches on an important topic - the development of composites that are safe for the environment. In this article, the authors study the thermal behavior of composites based on ethylene-norbornene copolymer filled with modified cellulose fibers.

I hope that the authors will take into account the following comments and make corrections.

Why did the authors choose this particular brand of pulp (Arbocel® UFC100)? What is its advantage? I recommend adding the characteristics of the object: indicate the degree of polymerization of cellulose, moisture content, alpha-cellulose content. Perhaps the “odorless” characteristic should be removed, since in this case the organoleptic properties are not determinative (line 97)

Line 97. Remove italics.

Line 97-98. The authors write that “it is in the form of a powder” and speak of “a fiber length of 8 microns”. It is better to choose one term - either talk about powder and particle size, or use only the concept of “fiber”

Line 103 "Vapor pressure of a compound is 0.2 hPa at 22 °C." In my opinion, this characteristic of maleic anhydride is not fundamental in this work, it is unnecessary

Table 1. For what purpose are the melting points of solvents shown in the table? It is enough to give only the boiling point.

Line 117 I recommend replacing the “composite” with the “copolymer”. “Composite” is used in relation to a filled system.

Line 157 Use the same style for weight percentages: replace “86 wt.%” with “86 wt%”

In Figure 2, I propose to show TGA curves for an unfilled copolymer, as well as data for an unmodified filler. Thus, the nature of the change in the thermal behavior of the composite, in comparison with pure materials, will be understood. (Some values ​​of mass loss temperatures for TOPAS are given in table 3 and they need to be reflected on the graph as well)

Figure 4, Figure 5. At least one of the figures should show DSC curves for ethylene-norbornene copolymer in order to evaluate the effect of filler on the thermal behavior of the composite (especially since the authors give some DSC data for the copolymer in Table 5)

Why on the DSC curves do the authors cite data only in the range up to 175 °C? Is there any data corresponding to the region of cellulose destruction (290-350 °С)?

Table 3. The temperatures given in the table for filled composites are very close in value; practically they lie within the error range. Authors should provide statistics. Is there a reproducibility of the indicated data for one sample when repeating the experiment 3-5 times?

In the article, it is advisable to provide the mechanical properties of the composites, as well as data on the morphology of the filled systems.

In my opinion, authors should strengthen the conclusion and provide potential applications for studied filled composites

Author Response

Institute of Polymer and Dye Technology

Technical University of Lodz

90-924 Lodz, ul Stefanowskiego 12/16, Poland

Tel.: +48 42 631 32 23, Fax: +48 42 636 25 43

February  28, 2020

Molecules

Dear Professor,

We are resubmitting our revised paper entitled Bio-inspired cellulose-filled polymer composites of an improved thermal performance (new title: Thermal behaviour of green cellulose-filled thermoplastic elastomer polymer composites) by, Stefan Cichosz, Anna Masek with a request to reconsider it for publication in Molecules.

We have carefully considered the Editor and Reviewers' comments. The manuscript was revised exactly according to these comments. The list of responses to the reviewer’s comments and corrections made in the manuscript is attached.

The manuscript has not been previously published, is not currently submitted for review to any other journal, and will not be submitted elsewhere before a decision is made by this journal.

For correspondence please use the following information:

corresponding author: Anna Masek

Institute of Polymer and Dye Technology

Technical University of Lodz

90-924 Lodz, ul Stefanowskiego 12/16, Poland

Tel.: +48 42 631 32 93

Fax: +48 42 636 25 43

e-mail: anna.masek@p.lodz.pl

Yours sincerely,

Ph. D., D.Sc. Anna Masek

All changes are marked with a green colour through whole manuscript.

Reviewer #1

The presented work is of scientific interest, and touches on an important topic - the development of composites that are safe for the environment. In this article, the authors study the thermal behavior of composites based on ethylene-norbornene copolymer filled with modified cellulose fibers. I hope that the authors will take into account the following comments and make corrections.

The comments are listed below:

  1. Why did the authors choose this particular brand of pulp (Arbocel® UFC100)? What is its advantage? I recommend adding the characteristics of the object: indicate the degree of polymerization of cellulose, moisture content, alpha-cellulose content. Perhaps the “odorless” characteristic should be removed, since in this case the organoleptic properties are not determinative (line 97).

Answer: We have chosen this material as it is a type of cellulose commonly used in industry, especially papermaking. Unfortunately, we are unable to indicate such information as a degree of polymerization and alpha-cellulose content. This is the thing we would like to establish in further research. We removed the “odorless” from the characteristics. What is more, moisture content is a subject of another study and it is very random depending on storage conditions.

  1. Line 97. Remove italics.

Answer: We are grateful for this comment. Italics are removed.

  1. Line 97-98. The authors write that “it is in the form of a powder” and speak of “a fiber length of 8 microns”. It is better to choose one term - either talk about powder and particle size, or use only the concept of “fiber”.

Answer: We are thankful for Reviewer’s advice. Mistake corrected.

  1. Line 103 "Vapor pressure of a compound is 0.2 hPa at 22 °C." In my opinion, this characteristic of maleic anhydride is not fundamental in this work, it is unnecessary.

Answer: This part has been removed.

  1. Table 1. For what purpose are the melting points of solvents shown in the table? It is enough to give only the boiling point.

Answer: Unnecessary information has been deleted.

  1. Line 117 I recommend replacing the “composite” with the “copolymer”. “Composite” is used in relation to a filled system.

Answer: We are sorry for this mistake. It has been corrected.

  1. Line 157 Use the same style for weight percentages: replace “86 wt.%” with “86 wt%”.

Answer: We are grateful for this comment. Mistake corrected.

  1. In Figure 2, I propose to show TGA curves for an unfilled copolymer, as well as data for an unmodified filler. Thus, the nature of the change in the thermal behavior of the composite, in comparison with pure materials, will be understood. (Some values ​​of mass loss temperatures for TOPAS are given in table 3 and they need to be reflected on the graph as well).

Answer: We are thankful for Reviewer’s advice. The graph has been improved.

  1. Figure 4, Figure 5. At least one of the figures should show DSC curves for ethylene-norbornene copolymer in order to evaluate the effect of filler on the thermal behavior of the composite (especially since the authors give some DSC data for the copolymer in Table 5) Why on the DSC curves do the authors cite data only in the range up to 175 °C? Is there any data corresponding to the region of cellulose destruction (290-350 °С)?

Answer: We are thankful for drawing our attention to this problem. The graphs has been improved. DSC experiment has been carried out mostly in order to see the changes in the material softening region. Therefore, we do not have data containing information about cellulose degradation in the composite sample. TGA experiment has been performed in order to observe composite sample degradation. Nevertheless, it is a valuable comment considering carrying out further research.

  1. Table 3. The temperatures given in the table for filled composites are very close in value; practically they lie within the error range. Authors should provide statistics. Is there a reproducibility of the indicated data for one sample when repeating the experiment 3-5 times?

Answer: We are grateful for this comment. Unfortunately, we are not able to carry out the further research on this material as it degraded and obtained values would be different. We would like to assure You that we have tested a few samples for three times at the beginning of carrying out measurements and the results were very close to each other. The reproducibility has been obtained within the same composite specimen.

  1. In the article, it is advisable to provide the mechanical properties of the composites, as well as data on the morphology of the filled systems. In my opinion, authors should strengthen the conclusion and provide potential applications for studied filled composites.

Answer: We are thankful for Reviewer’s advice. The mechanical properties of the composite and the data considering the morphology is a subject of another article which is now under review in another scientific journal. Conclusion section has been improved.

Reviewer 2 Report

The paper presents a series of data on the thermal stability and thermal properties of elastomeric TOPAS filled with cellulose. The presence of cellulose based filler should improve the thermal performance of the polymer composites, as reported in the title.

The author set up significant numbers of experiments of modified cellulosa fibers evaluating the effect on the thermal performance of Topas composites.

However, it is difficult to rationalize the results presented because differences in the thermal properties values measured are low. Moreover, the treatments that improve some properties (eg thermal stability) do not affect other properties (eg glass transition) and vice versa. The authors should justify some data significantly different from the others, such as the data with Ea1 230 kJ / mol presented in Table 4.

Author Response

Institute of Polymer and Dye Technology

Technical University of Lodz

90-924 Lodz, ul Stefanowskiego 12/16, Poland

Tel.: +48 42 631 32 23, Fax: +48 42 636 25 43

February  28, 2020

Molecules

Dear Professor,

We are resubmitting our revised paper entitled Bio-inspired cellulose-filled polymer composites of an improved thermal performance (new title: Thermal behaviour of green cellulose-filled thermoplastic elastomer polymer composites) by, Stefan Cichosz, Anna Masek with a request to reconsider it for publication in Molecules.

We have carefully considered the Editor and Reviewers' comments. The manuscript was revised exactly according to these comments. The list of responses to the reviewer’s comments and corrections made in the manuscript is attached.

The manuscript has not been previously published, is not currently submitted for review to any other journal, and will not be submitted elsewhere before a decision is made by this journal.

For correspondence please use the following information:

corresponding author: Anna Masek

Institute of Polymer and Dye Technology

Technical University of Lodz

90-924 Lodz, ul Stefanowskiego 12/16, Poland

Tel.: +48 42 631 32 93

Fax: +48 42 636 25 43

e-mail: anna.masek@p.lodz.pl

Yours sincerely,

Ph. D., D.Sc. Anna Masek

All changes are marked with a green colour through whole manuscript.

Reviewer #2

The paper presents a series of data on the thermal stability and thermal properties of elastomeric TOPAS filled with cellulose. The presence of cellulose based filler should improve the thermal performance of the polymer composites, as reported in the title. The author set up significant numbers of experiments of modified cellulosa fibers evaluating the effect on the thermal performance of Topas composites.

The comments are listed below:

  1. However, it is difficult to rationalize the results presented because differences in the thermal properties values measured are low. Moreover, the treatments that improve some properties (eg thermal stability) do not affect other properties (eg glass transition) and vice versa. The authors should justify some data significantly different from the others, such as the data with Ea1 230 kJ / mol presented in Table 4.

Answer: We are thankful for drawing our attention to this problem. Yet, considering natural-filled polymer composite samples an improvement in thermal stability even of a few degrees is a significant change. Moreover, it has to be taken into consideration that actually ethylene segment glass transition temperature is highly altered by the type of performed modification. All presented information was revised and elevated once more.

Reviewer 3 Report

The formation of new materials based on the ethylene-norbornene copolymer and cellulose seems to be an interesting concept since cellulose essentially is a biodegradable material. The presentation of results, however, needs to be slightly improved and the specimen requires description in more detail:

  1. Which of the cellulose modification methods (table 2) was applied while the polymer composite samples were being prepared?
  2. The composition and abbreviations pertaining to the obtained materials should be presented in the form of a table.
  3. The scientific novelty should be emphasized.
  4. The Authors should have discussed the effect of the fibers on the surface energy of Topas-based composites.
  5. The Authors should have paid more attention to the conclusions in aim to summarize more clearly the obtained results,

Taking into account all of the comments I suggest a minor revision of the submitted manuscript.

Author Response

Institute of Polymer and Dye Technology

Technical University of Lodz

90-924 Lodz, ul Stefanowskiego 12/16, Poland

Tel.: +48 42 631 32 23, Fax: +48 42 636 25 43

February  28, 2020

Molecules

Dear Professor,

We are resubmitting our revised paper entitled Bio-inspired cellulose-filled polymer composites of an improved thermal performance (new title: Thermal behaviour of green cellulose-filled thermoplastic elastomer polymer composites) by, Stefan Cichosz, Anna Masek with a request to reconsider it for publication in Molecules.

We have carefully considered the Editor and Reviewers' comments. The manuscript was revised exactly according to these comments. The list of responses to the reviewer’s comments and corrections made in the manuscript is attached.

The manuscript has not been previously published, is not currently submitted for review to any other journal, and will not be submitted elsewhere before a decision is made by this journal.

For correspondence please use the following information:

corresponding author: Anna Masek

Institute of Polymer and Dye Technology

Technical University of Lodz

90-924 Lodz, ul Stefanowskiego 12/16, Poland

Tel.: +48 42 631 32 93

Fax: +48 42 636 25 43

e-mail: anna.masek@p.lodz.pl

Yours sincerely,

Ph. D., D.Sc. Anna Masek

All changes are marked with a green colour through whole manuscript.

Reviewer #3

The formation of new materials based on the ethylene-norbornene copolymer and cellulose seems to be an interesting concept since cellulose essentially is a biodegradable material. The presentation of results, however, needs to be slightly improved and the specimen requires description in more detail.

The comments are listed below:

  1. Which of the cellulose modification methods (table 2) was applied while the polymer composite samples were being prepared?

Answer: All of the modifications has been performed. Cellulose fibres were treated according to each modification method presented in Table 2 and then implemented into the polymer matrix in order to create a composite material.

  1. The composition and abbreviations pertaining to the obtained materials should be presented in the form of a table.

Answer: We are grateful for this advice. Abbreviations are explained in Table 2 and the amount of cellulose added to the polymer matrix was always the same which is written in 2.1.3. section.

  1. The scientific novelty should be emphasized.

Answer: We are thankful for drawing our attention to this problem. The novelty has been described more precisely.

  1. The Authors should have discussed the effect of the fibers on the surface energy of Topas-based composites.

Answer: The description of surface free energy changes has been revised and improved.

  1. The Authors should have paid more attention to the conclusions in aim to summarize more clearly the obtained results.

Answer: We are grateful for this comment. Conclusions has been revised and improved.

Reviewer 4 Report

This manuscript reports the results of an evaluation of the impact of the presence of modified cellulose on the thermal stability of an ethylene/norbornene copolymer. The compositions are referred to as composites but no evidence for composite formation is presented. Was interfacial adhesion measured? Were mechanical properties improved by the presence of the modified cellulose? There is confusion between "filler" and "reinforcing agent". There is a world of difference between simple filled polymers and polymer composites. This should be clarified. It should be noted that activation energies obtained using variable temperature techniques represent global values reflecting contributions from all processes occurring which change as a function of temperature [see J. Therm. Anal. Calorim., 2008, 93, 27]. This should be noted. The degradation processes in the presence of modified cellulose are likely different from those for the polymer alone.

It has been demonstrated that the thermal stability for the polymer/modified cellulose blend is lower than that for the polymer alone. This is touted as a positive for waste management. However, the propensity for thermal degradation is probably not reflective of biodegradability. A discussion of any relationship between the two should be added or the reference to waste management removed.

The manuscript will need significant revision for accuracy, clarity and readability. Corrections are penciled-in directly on pages of the manuscript attached. These are indicative of the kinds of changes needed throughout. Author's names and et. al. should be omitted.

The title is somewhat misleading. Cellulose is not "bioinspired", it is a biopolymer. The thermal performance is not "improved". The thermal stability of the blends is actually lower than that for the polymer alone.

Author Response

Institute of Polymer and Dye Technology

Technical University of Lodz

90-924 Lodz, ul Stefanowskiego 12/16, Poland

Tel.: +48 42 631 32 23, Fax: +48 42 636 25 43

February  28, 2020

Molecules

Dear Professor,

We are resubmitting our revised paper entitled Bio-inspired cellulose-filled polymer composites of an improved thermal performance (new title: Thermal behaviour of green cellulose-filled thermoplastic elastomer polymer composites) by, Stefan Cichosz, Anna Masek with a request to reconsider it for publication in Molecules.

We have carefully considered the Editor and Reviewers' comments. The manuscript was revised exactly according to these comments. The list of responses to the reviewer’s comments and corrections made in the manuscript is attached.

The manuscript has not been previously published, is not currently submitted for review to any other journal, and will not be submitted elsewhere before a decision is made by this journal.

For correspondence please use the following information:

corresponding author: Anna Masek

Institute of Polymer and Dye Technology

Technical University of Lodz

90-924 Lodz, ul Stefanowskiego 12/16, Poland

Tel.: +48 42 631 32 93

Fax: +48 42 636 25 43

e-mail: anna.masek@p.lodz.pl

Yours sincerely,

Ph. D., D.Sc. Anna Masek

Reviewer #4

This manuscript reports the results of an evaluation of the impact of the presence of modified cellulose on the thermal stability of an ethylene/norbornene copolymer.

The comments are listed below:

  1. The compositions are referred to as composites but no evidence for composite formation is presented. Was interfacial adhesion measured? Were mechanical properties improved by the presence of the modified cellulose? There is confusion between "filler" and "reinforcing agent". There is a world of difference between simple filled polymers and polymer composites. This should be clarified.

Answer: We are grateful for drawing our attention to this problem. Whole manuscript has been revised considering these misleading terms. The mechanical properties of the composite and the data considering the morphology is a subject of another article which is now under review in another scientific journal.

  1. The degradation processes in the presence of modified cellulose are likely different from those for the polymer alone. It has been demonstrated that the thermal stability for the polymer/modified cellulose blend is lower than that for the polymer alone. This is touted as a positive for waste management. However, the propensity for thermal degradation is probably not reflective of biodegradability. A discussion of any relationship between the two should be added or the reference to waste management removed. The manuscript will need significant revision for accuracy, clarity and readability. Corrections are penciled-in directly on pages of the manuscript attached. These are indicative of the kinds of changes needed throughout. Author's names and et. al. should be omitted.

Answer: We are thankful for this comment and for preparing such adequate corrections. Whole manuscript has been revised once more and mistakes are corrected. Degradation of polymer composites is more connected with increased polar part of surface free energy and easier wetting of specimens. Thermal decomposition is assessed thanks to DSC and TGA. Information added in conclusions.

  1. The title is somewhat misleading. Cellulose is not "bioinspired", it is a biopolymer. The thermal performance is not "improved". The thermal stability of the blends is actually lower than that for the polymer alone.

Answer: This is a very valuable comment which we are grateful for. The title has been changed: Thermal behaviour of green cellulose-filled thermoplastic elastomer polymer composites.

Round 2

Reviewer 1 Report

A reply «The graph has been improved» was received to the comment «In Figure 2, I propose to show TGA curves for an unfilled copolymer, as well as data for an unmodified filler. Thus, the nature of the change in the thermal behavior of the composite, in comparison with pure materials, will be understood».

Unfortunately, in the manuscript file, I did not find any changes in Figure 2. Perhaps this is a technical error. I gently recommend that the authors show in Figure 2 the TGA curves for the unfilled copolymer.

The authors of the work set the following task: "The aim of the performed modification process was to reduce the moisture content in the cellulose fibers, ..." (line 83-84). However, the authors do not give the characteristics of the introduced cellulose fibers, in particular, do not indicate the moisture content. In my opinion, this is an important component necessary for understanding the thermal behavior of filled systems in the context of this work.

Author Response

Institute of Polymer and Dye Technology

Technical University of Lodz

90-924 Lodz, ul Stefanowskiego 12/16, Poland

Tel.: +48 42 631 32 23, Fax: +48 42 636 25 43

March 09, 2020

Molecules

Dear Professor,

We are resubmitting our revised   

Reviewer #1

The comments are listed below:

  1. A reply «The graph has been improved» was received to the comment «In Figure 2, I propose to show TGA curves for an unfilled copolymer, as well as data for an unmodified filler. Thus, the nature of the change in the thermal behavior of the composite, in comparison with pure materials, will be understood». Unfortunately, in the manuscript file, I did not find any changes in Figure 2. Perhaps this is a technical error. I gently recommend that the authors show in Figure 2 the TGA curves for the unfilled copolymer.

Answer: We are sorry for this mistake. The figures has been corrected in the manuscript.

  1. The authors of the work set the following task: "The aim of the performed modification process was to reduce the moisture content in the cellulose fibers, ..." (line 83-84). However, the authors do not give the characteristics of the introduced cellulose fibers, in particular, do not indicate the moisture content. In my opinion, this is an important component necessary for understanding the thermal behavior of filled systems in the context of this work.

Answer: We have added the following information about the cellulose properties: The whole modification process and its influence on the cellulose properties is widely described in the previous article concentrating on this topic [38] and the short summary is presented in Table 2. It is a topic of a separate article.

Reviewer 2 Report

The authors considered the reviewer's suggestions and also improved the text by filling in the gaps and better explaining the experimental data obtained.

On the basis of these considerations, the article can be accepted in this form.

Author Response

Institute of Polymer and Dye Technology

Technical University of Lodz

90-924 Lodz, ul Stefanowskiego 12/16, Poland

Tel.: +48 42 631 32 23, Fax: +48 42 636 25 43

March 09, 2020

Molecules

Dear Professor,

We are resubmitting our revised paper entitled Bio-inspired cellulose-filled polymer composites of an improved thermal performance (new title: Thermal behaviour of green cellulose-filled thermoplastic elastomer polymer composites) by, Stefan Cichosz, Anna Masek with a request to reconsider it for publication in Molecules.

We have carefully considered the Editor and Reviewers' comments. The manuscript was revised exactly according to these comments. The list of responses to the reviewer’s comments and corrections made in the manuscript is attached.

The manuscript has not been previously published, is not currently submitted for review to any other journal, and will not be submitted elsewhere before a decision is made by this journal.

For correspondence please use the following information:

corresponding author: Anna Masek

Institute of Polymer and Dye Technology

Technical University of Lodz

90-924 Lodz, ul Stefanowskiego 12/16, Poland

Tel.: +48 42 631 32 93

Fax: +48 42 636 25 43

e-mail: anna.masek@p.lodz.pl

Yours sincerely,

Ph. D., D.Sc. Anna Masek

Reviewer 4 Report

This manuscript is improved. Yet, no evidence for composite formation has been provided (it is suggested that this is being generated and will be presented in a forthcoming publication). At line 164, a couple of sentences could be added to reflect an increase in mechanical properties upon incorporation of the fibers, i.e., that genuine composites are being formed. If this is not possible, these should be referred to as "blends" rather than "composites".

"Activation energy" should be "apparent activation energy" throughout. The "activation energies" obtained using variable temperature techniques represent global values with contributions from all processes occurring which change as a function of temperature [see J. Therm. Anal. Calorim., 2008, 93, 27.].

Author Response

Institute of Polymer and Dye Technology

Technical University of Lodz

90-924 Lodz, ul Stefanowskiego 12/16, Poland

Tel.: +48 42 631 32 23, Fax: +48 42 636 25 43

March 09, 2020

Molecules

Dear Professor,

We are resubmitting our revised paper entitled Bio-inspired cellulose-filled polymer composites of an improved thermal performance (new title: Thermal behaviour of green cellulose-filled thermoplastic elastomer polymer composites) by, Stefan Cichosz, Anna Masek with a request to reconsider it for publication in Molecules.

We have carefully considered the Editor and Reviewers' comments. The manuscript was revised exactly according to these comments. The list of responses to the reviewer’s comments and corrections made in the manuscript is attached.

The manuscript has not been previously published, is not currently submitted for review to any other journal, and will not be submitted elsewhere before a decision is made by this journal.

For correspondence please use the following information:

corresponding author: Anna Masek

Institute of Polymer and Dye Technology

Technical University of Lodz

90-924 Lodz, ul Stefanowskiego 12/16, Poland

Tel.: +48 42 631 32 93

Fax: +48 42 636 25 43

e-mail: anna.masek@p.lodz.pl

Yours sincerely,

Ph. D., D.Sc. Anna Masek

Reviewer #4

The comments are listed below:

  1. This manuscript is improved. Yet, no evidence for composite formation has been provided (it is suggested that this is being generated and will be presented in a forthcoming publication). At line 164, a couple of sentences could be added to reflect an increase in mechanical properties upon incorporation of the fibers, i.e., that genuine composites are being formed. If this is not possible, these should be referred to as "blends" rather than "composites".

Answer: We are grateful for this suggestion. As we would like to be precise, the material is referred as “blend” throughout the whole manuscript. Further results are going to be a subject of another research being under review. Therefore, we are not able to double the presented data. It would be inappropriate and against of the rules respected by journals.

  1. "Activation energy" should be "apparent activation energy" throughout. The "activation energies" obtained using variable temperature techniques represent global values with contributions from all processes occurring which change as a function of temperature [see J. Therm. Anal. Calorim., 2008, 93, 27.].

Answer: We are thankful for this comment and the mistake has been corrected.